# Day Ahead Electric Load Forecast: A Comprehensive LSTM-EMD Methodology and Several Diverse Case Studies

Michael Wood [1,2,*], Emanuele Ogliari [1], Alfredo Nespoli [1], Travis Simpkins [2] and Sonia Leva [1]

1 Department of Energy, Politecnico di Milano, Via Lambruschini 4a, 20156 Milan, Italy
2 muGrid Analyics LLC., 14143 Denver West Parkway, Suite 100, Golden, CO 80433, USA
* Correspondence: michael.wood@polimi.it

**Abstract:** Optimal behind-the-meter energy management often requires a day-ahead electric load forecast capable of learning non-linear and non-stationary patterns, due to the spatial disaggregation of loads and concept drift associated with time-varying physics and behavior. There are many promising machine learning techniques in the literature, but black box models lack explainability and therefore confidence in the models' robustness can't be achieved without thorough testing on data sets with varying and representative statistical properties. Therefore this work adopts and builds on some of the highest-performing load forecasting tools in the literature, which are Long Short-Term Memory recurrent networks, Empirical Mode Decomposition for feature engineering, and k-means clustering for outlier detection, and tests a combined methodology on seven different load data sets from six different load sectors. Forecast test set results are benchmarked against a seasonal naive model and SARIMA. The resultant skill scores range from −6.3% to 73%, indicating that the methodology adopted is often but not exclusively effective relative to the benchmarks.

**Keywords:** electric load; forecasting; neural networks; LSTM; EMD; industrial; commercial

## 1. Introduction

The renewable energy transition will necessarily change how we plan, operate, and study electricity networks. In their public policy work, researchers in [1] conclude that the US electric grid can reach a 90% clean case by 2035 without new coal or gas plants and while reducing costs. For regions without substantial hydroelectric or nuclear, this could mean a grid almost entirely balanced by intermittent renewable sources, storage, and demand response. Much of that capacity will be centralized plants with significant economies of scale, however, distributed resources like solar photovoltaic (PV) and battery storage have additional benefits. These include closely coupling load with generation, resilient backup power [2], leveraging privately held capital, much-needed elasticity in market demand curves [3], and a spatial peer effect [4] where consumers who commonly see distributed energy projects appear to engage more in energy issues.

An important regime of distributed energy is Behind-the-Meter (BTM) energy because of the direct coupling with load and the presence of a consumer meter and tariffs. Compared to a merchant PV plant or battery system operated by a distribution company, BTM energy optimization has a common formulation that depends strongly on the available energy resources and the tariff structure. A non-linear unit commitment problem must then be solved during the resource planning phase and continually during operation [5]. A common strategy is to forecast load and, if applicable, renewable generation. Then a mixed-integer linear program is formulated according to the energy resources and tariff and solved to determine the dispatch strategy such as in [6]. Solar PV forecasting is well investigated and summarized by [7,8], as is formulating and solving the optimization problem.

The problem considered by this work is that most research into non-system load forecasting considers only one of the four load sectors: residential, commercial, industrial,

or transport. Table 1 summarizes a small review of the relevant published works and the data source by load sector. Only two papers span multiple sectors, although [9] includes 117 different building types in the commercial sector. Usually, there is little analysis of the physical phenomena or statistical properties of the data. And since the publicly available data are usually pre-processed, often few outliers or bad data are present which can drastically skew forecast model training. Authors sometimes indicate these shortcomings and cite a lack of available data. The result is lower confidence that the proposed methods will generalize to new datasets, which is the classic problem of black box machine learning techniques.

**Table 1.** Electric Load Sectors in Forecast Literature.

| Sector | Papers (Single Site) | Papers (Multiple Sites) |
|---|---|---|
| Residential | [10–15] | [16–20] |
| Commercial | [10,21,22] | [9,16] |
| Industrial | [10,23–25] | [16] |
| All Three | [10] | [16] |

A need exists to apply a high-accuracy load forecasting technique to several datasets from different load sectors. The technique should be combined into a single methodology along with pre-processing for robustness to outliers and bad data. The methodology must be completely data-driven and should not depend on exogenous features which are not necessarily available. The benchmark forecast models should include seasonal Naive Persistence (NP), because it is transparent and simple to characterize, and Seasonal Autoregression Integrated Moving Average (SARIMA) because it is fairly effective for univariate time series and can quickly and transparently be trained with auto ARIMA packages in R and Python.

In the literature, electric load forecasts are traditionally studied at the transmission network or substation scale [26]. Recently residential forecasts are of increased interest for flexibility in smart grids [12] and microgrids [27], and for optimization at the consumer level [28]. Individual load equipment monitoring is recently made more feasible by non-intrusive load disaggregation as in [29]. Electric Vehicle (EV) charging stations are an important area of study because of the expected demand growth and the opportunities around charging and vehicle-to-grid services [30].

Long-term (>3 years) load and renewable generation forecasts allow operators to plan infrastructure upgrades and inform generation investors to estimate revenues through the project lifetime [31]. Medium-term (0.5–36 months) forecasts give information to forward market participants and to regulators who often require seasonal variations in available generating capacity. Short-term (1–14 days) forecasts inform bidding strategies in the spot markets and aid network operators in managing power flows. Nowcasting (15–60 min) is a relatively new time scale that applies especially to predicting large changes in solar or wind production [32] and is useful in various operational environments.

Methods to forecast load include physical building modeling, statistical regression models, and machine learning. Physical modeling tools such as DOE-2 and EnergyPlus describe the physical building processes of heating, cooling, ventilation, and lighting while estimating the remaining electrical load with different techniques. These methods are often the least accurate but don't require any load time series measurements, so can be used during the building planning phase [33]. Statistical regression models such as SARIMA and exponential smoothing build upon the linear regression concept, incorporating differencing, a moving average, seasonality, exogenous variable, and non-linearity [33]. However there is good evidence that statistical regression models are inadequate for highly non-linear, non-Gaussian, and often non-stationary time series [34]. Machine learning algorithms such as Support Vector Machines [35], Radial Basis Functions [36], Artificial Neural Network (ANN) networks [15], and Long Short-Term Memory (LSTM) networks are performing well in sequential data processing problems such as load forecasting [37], public transportation

flows [38], and natural language processing [39]. LSTM appears particularly promising for small-scale disaggregated resiential forecasts [37], and modifications are being investigated such as sequence-to-sequence structures [15], ensemble bagging methods [25], and EMD feature extraction [40]. These works typically focus on model development and comparison and only test on a small amount of data from one load sector. Therefore the strength of the conclusions is moderate at best. Certain pre-processing approaches can enhance feature recognition using clustering such as in [41,42], achieve dimensionality reduction as with Principal Component Analysis [42] or otherwise decompose the signal such as Empirical Mode Decomposition (EMD), investigated recently in the literature by [35,38]. With machine learning approaches, large training datasets can be especially effective at reducing forecast error [43] even for relatively simple models [44].

The novelties of this work are (1) the development of a high-performing and comprehensive load forecasting methodology that is robust to outliers and bad data, and (2) the application of the methodology to load data in each other following sectors: residential, commercial, industrial, and transport. The rest of the paper is organized as follows:

1. Methodology: Data pre-processing, model training and evaluation, and operational forecasting
2. Case Studies: Several datasets are studied via the methodology including buildings, an electric vehicle charging station, and transmission networks
3. Results: The intermediate and final results are evaluated and contextualized
4. Conclusions: A final summary and evaluation of the methodology are discussed

## 2. Methodology

The proposed methodology in Figure 1 builds on and improves several other techniques in the literature. The goal is to predict day-ahead electric load given nothing but a historical load time series. The predictions are exactly at the 24-h horizon, with the same time interval as the historical series. The Hotel 1 dataset (in Table 2) was the basis for developing this methodology, which is 3.2 years (after pre-processing) of 15-min interval measurements, with a maximum of 1.76 MW. The other datasets are used to test the ability of the methodology to generalize to other load sectors, including a comparison to power system loads.

**Table 2.** Description of Datasets.

| Site | Load Sector [1] | Koppen Climate | Length [y] | Interval [min] | Peak [MW] | Autocorr. 1/7 day | ADF Statistic |
|---|---|---|---|---|---|---|---|
| Residence [2] | Res. | BSk | 16.4 | 60 | 0.007 | 0.70/0.63 | −17.1 |
| Hotel 1 | Com. | Af | 3.2 | 15 | 1.76 | 0.95/0.92 | −4.9 |
| Hotel 2 | Com. | BSk | 2.0 | 15 | 0.45 | 0.87/0.80 | −5.5 |
| Manufacturing Plant | Ind. | Dfa | 2.9 | 15 | 13.0 | 0.45/0.93 | −20.1 |
| EV Charging Station [3] | Tra. | Cfa | 2.7 | 10 | 0.16 | 0.70/0.84 | −17.4 |
| Distribution Network | Sys. | BSk | 13.8 | 60 | 1.9 | 0.94/0.86 | −10.5 |
| Transmission Network [4] | Sys. | Many | 10.0 | 60 | 59,700 | 0.78/0.89 | −17.7 |

[1] Residential, Commercial, Industrial, Transport, System. [2] NREL Habitat for Humanity Zero Energy Home, USA. [3] Caltech Adaptive Charging Network, USA. [4] TERNA, Italy.

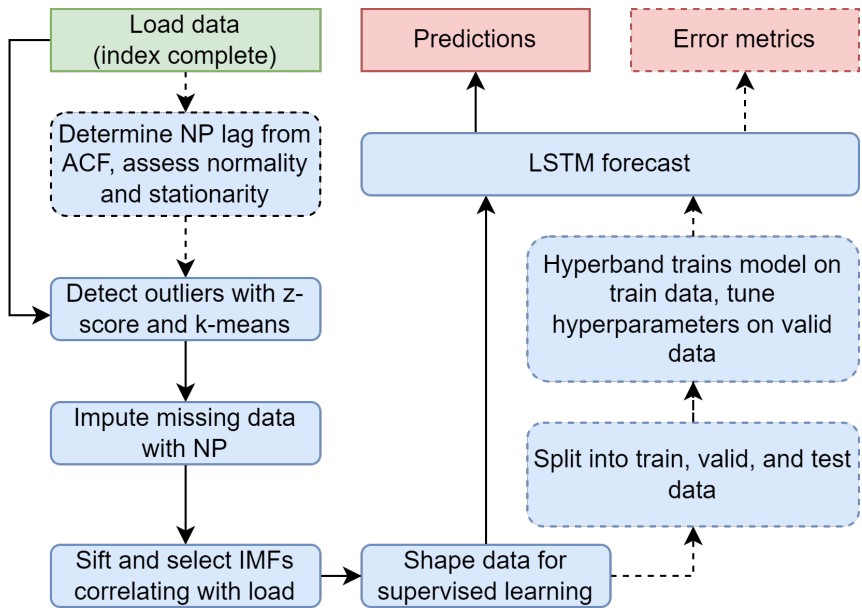

**Figure 1.** A simplified process diagram of the long short-term memory (LSTM) methodology begins with the input cumulative raw data and ends at the output of predicted load values and error metrics. For a new dataset, all blocks are included and dashed lines are prioritized over solid ones, such that naive persistence (NP) lag and the autocorrelation function (ACF) are only established once. The dashed lines and blocks are omitted after data have been pre-processed and the model has been hyperparameter optimized, such as during an operational forecast.

### 2.1. Error Metrics

Error metrics must be carefully chosen for adequate model training, statistical properties, and comparability with other results in the literature. A robust training metric is Mean Squared Error (MSE), which is symmetrical and penalizes larger error magnitudes unless there are many outliers [45]. For comparison with results in the literature Root Mean Squared Error (RMSE) is presented, having similar properties as MSE although with the same units as the endogenous data. Furthermore, in this work RMSE is used to calculate skill score (SS), which is a helpful indicator of performance relative to a benchmark model, such as NP or SARIMA. The error metrics above are calculated as:

$$\text{MSE} = \frac{1}{N} \sum_{t=1}^{N} (y_t - \hat{y}_t)^2 \tag{1}$$

$$\text{RMSE} = \sqrt{\text{MSE}} \tag{2}$$

$$\text{SS} = 1 - \frac{\text{RMSE}}{\text{RMSE}_{benchmark}} \tag{3}$$

where $N$ is the number of samples, $y_t$ is the measured value at time $t$, $\hat{y}_t$ is the estimated value at time $t$, and $\text{RMSE}_{\text{benchmark}}$ is the RMSE of benchmark model.

In this work, the seasonal lag period is either one day or seven days, depending on which yields a lower RMSE on training data. Load data with weekend and holiday concept drift should benefit from choosing a seven-day lag period rather than one day.

### 2.2. Data Pre-Processing

Raw data are pre-processed in a modified Box-Jenkins method to assess seasonality, remove outliers to improve model training and discern characteristics of the data useful for interpreting the results. First, the date-time index is made complete (no missing indices and a constant interval) and invalid data such as NaN are replaced with simple naive persistence if possible.

Normality is assessed using the Shapiro-Wilk regression method since it is effective over a wide range of asymmetric distributions [46]. The entire data set is assessed, as well as the 168-hour-weekday subsets of the data where the first subset is every Monday from 0:00 to 1:00, the second is every Monday from 1:00 to 2:00, and so on. Whereas the whole data set is typically not Gaussian (owing partially to the sinusoidal, diurnal nature of electric load) many of the 168 subsets often are.

Stationarity is typically a large hurdle for autoregression [47] and neural network forecasts [48]. Here, it is assessed using the Augmented Dicky-Fuller (ADF) test for the presence of a unit root (such that $\gamma$ is zero) in the following non-linear model:

$$\Delta y_t = \alpha + \beta t + \gamma y_{t-1} + \delta_1 \Delta y_{t-1} + \delta_2 \Delta y_{t-2} + \ ... \tag{4}$$

where $y_t$ is the value at time $t$ and the coefficients are $\alpha$, $\beta$, $\gamma$, etc. The $\delta$ coefficients of a stationary series should be non-zero, indicating that previous differences ($\Delta y$) have information about the next difference. This is because a stationary series must return to a constant mean, where large positive differences tend to be followed by large negative differences.

Outliers are first labeled based on a z-score greater than three for each of the 168-hour-weekday subsets, calculated as:

$$z_{i,h} = \frac{|y_{i,h} - \bar{y}_h|}{\sigma_h} \tag{5}$$

where $h$ is the hour of the day (0 to 23), $y_{i,h}$ is the $i$th value at hour $h$, $\bar{y}_h$ is the mean for hour $h$, and $\sigma_h$ is the standard deviation for hour $h$. Since many of the hour-weekday subsets are highly non-Gaussian, k-means clustering is executed with the original $k$ guess as equal to the number of outliers labeled by z-score. In this way values that are geometrically close to z-score outliers are labeled, even if they are less than three standard deviations from the mean. K-means clustering iteratively assigns all points to one of $k$ geometric clusters, calculates the centroid of the points, then redefines clusters based on the centroids, and stops when some criterion is met, such that:

$$min\left(\sum_{j=1}^{k} \sum_{i=1}^{n} ||x_i^{(j)} - c_j||^2\right) \tag{6}$$

where $k$ is the numbers of clusters, $c_j$ is the centroid of cluster $j$, $n$ is the number of values in cluster $j$, and $x_i^{(j)}$ is the $i$th value of cluster $j$.

Once labeled, outliers are replaced with an immediate naive persistence value if available. If not (in the case of many series outliers) the seasonal naive persistence is taken as the replacement value, where the seasonal lag is either one or seven days. The seasonality of the data set is established using autocorrelation.

Finally, all data are min-max scaled, to not bias the neural network high or low and to not saturate the sigmoid activation function.

### 2.3. Feature Engineering

Pearson correlation with the endogenous load is used to quickly separate useful features from those that will unnecessarily slow down training. Electric consumption is known to correlate with exogenous features like temperature and calendar effects [49], but any meteorological data must be separately obtained in real-time, which may not be possible or economic. Therefore methods that utilize only the endogenous load data should be evaluated first for an operational forecast at the building scale.

This methodology includes calendar information such as day-of-year, day-of-week, and hour-of-day as a triangle-shaped feature. For example, the triangle begins on 1 January with a value of 0.0, peaks on 2 July with a value of 1.0, and ends on 31 December with a value of 0.0. Sinusoid or radial basis function signals could be used instead of triangular signals, but the daily and annual IMFs in practice already take a sinusoid shape.

Empirical Mode Decomposition

Deep learning algorithms have a documented ability to learn complex patterns [50], but these large models can be difficult to train and computationally expensive to run and re-train. For distributed energy projects, multiple forecasts and a linear program solver may need to run on edge computing devices rather than powerful cloud or traditional servers. By using very different mathematical techniques than neural networks signal processing methods can efficiently aid feature recognition. EMD is a fast and adaptive tool giving immediate frequency information on the local characteristics of the signal based on the Hilbert-Huang transform, also for non-linear and non-stationary series [51]. The result of EMD is a small number of Intrinsic Mode Functions (IMFs) which algebraically sum to the original signal. For example, in Figure 2 a synthetic signal, created by summing a sinusoid and triangle signal, is decomposed into two IMFs and a residual which are the same as the original sinusoid and triangle signals.

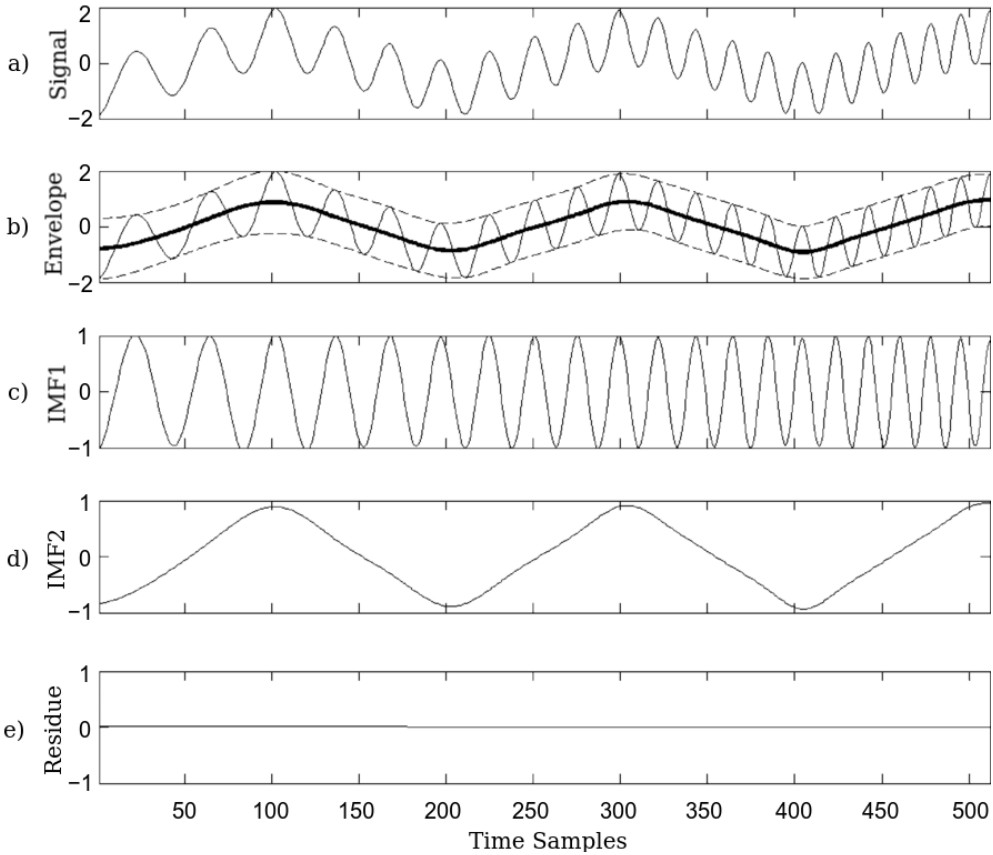

**Figure 2.** Example [52] of EMD on a synthetic signal (**a**), which was constructed by summing a sinusoid and triangle signal. Quartic splines are calculated to form the dotted-line envelope in (**b**). Then first recursive EMD sifting process produces the first IMF in (**c**), and then afterwards a second in (**d**). Remaining is the residual in (**e**). In this example the residual is exactly zero so the decomposition is perfect.

The EMD method interpolates consecutive local maxima of the time series with quartic splines, and consecutive local minima of the time series with quartic splines This creates an envelope in which lay all the points of the time series [51]. For each sample, the mean of the top and bottom envelopes are calculated. This mean is subtracted from the time series, and the process is repeated recursively until the result matches the criteria of an IMF. This first IMF is subtracted from the original time series and the process repeats, creating numerous IMFs until the last residual has less than two maxima or minima [51].

*2.4. Long Short-Term Memory Model*

Recurrent neural networks such as LSTM add a recursive information path to the neural cell, creating a limited capacity for outputs to be affected by past outputs, which functions as a kind of short-term memory. This can be quite effective for processing sequences of data [53]. Multiple features are cleanly handled with a second dimension of the input data structure, rather than concatenating along the same axis as the time series, as is the case with ANNs. Also, recurrent networks can inherently process sequences of different lengths. The basic recurrent network structure is shown in Figure 3. Some resources show or imply a concatenation procedure where $\mathbf{h}_{t-1}$ and $\mathbf{x}_{(t)}$ enter the fully connected layers, which is unknown to the authors and not implemented in the Tensorflow library. If $\mathbf{h}_{t-1}$ and $\mathbf{x}_{(t)}$ were concatenated there would only be a single weight matrix for the new concatenated vector, rather $\mathbf{h}_{t-1}$ and $\mathbf{x}_{(t)}$ each have a distinct weight matrix and are then summed together along with the bias vector.

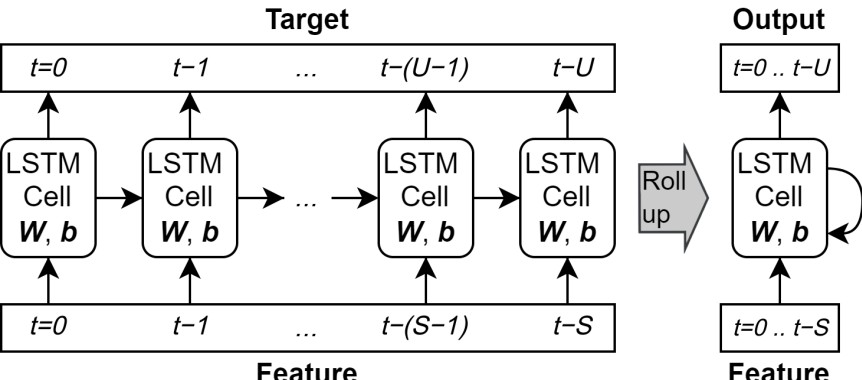

**Figure 3.** The feature sequence (length *S*) is input to the LSTM cell chronologically, up to the most recent value at $t = 0$, and can be depicted "unrolled" (left) with the same cell multiple times, or "rolled up" showing one cell and the recurrent loop. In either case there is only one set of **W**-weights ($\mathbf{W}_{xi}$, $\mathbf{W}_{hi}$, etc) and **b**-biases ($\mathbf{b}_i$, $\mathbf{b}_f$, etc). Between time steps, the LSTM cell passes **c** and **h** to itself in the next time step.

However, the recurrent neural cell's internal state is prone to changing quickly after just a few inputs, so the memory function is short-term. Also, the back-propagation error gradients are prone to vanishing or exploding, even with the use of a non-saturating activation function [45]. The LSTM cell mitigates this problem by adding a forget gate which allows the cell to selectively forget and remember information trying to update the internal state [54]. This limited memory may be effective for hundreds of samples, but likely not thousands.

In Figure 4 [45] the LSTM cell components and signals are shown. The new candidate for cell state $\mathbf{g}_{(t)}$ is the result of a fully connected neural layer with new input data $\mathbf{x}_{(t)}$ and the last hidden state $\mathbf{h}_{(t-1)}$. The flow of information inside the cell is controlled by gates including the forget gate $\mathbf{f}_{(t)}$, input gate $\mathbf{i}_{(t)}$, and output gate $\mathbf{o}_{(t)}$. The cell output is $\mathbf{y}_{(t)}$ [45].

This structure allows the cell to selectively maintain information in the cell state $\mathbf{c}_{(t)}$ based on its usefulness in decreasing model loss. In this work only LSTM models of three or four layers are used: an input layer, one or two hidden LSTM layers, and a fully connected output layer. For models in this methodology with two LSTM layers, the layers are sequence-to-sequence connected.

$$\mathbf{i}_{(t)} \quad = \sigma\left(\mathbf{W}_{xi}^{\top}\mathbf{x}_{(t)} + \mathbf{W}_{hi}^{\top}\mathbf{h}_{(t-1)} + \mathbf{b}_i\right) \tag{7}$$

$$\mathbf{f}_{(t)} \quad = \sigma\left(\mathbf{W}_{xf}^{\top}\mathbf{x}_{(t)} + \mathbf{W}_{hf}^{\top}\mathbf{h}_{(t-1)} + \mathbf{b}_f\right) \tag{8}$$

$$\mathbf{o}_{(t)} \quad = \sigma\left(\mathbf{W}_{xo}^{\top}\mathbf{x}_{(t)} + \mathbf{W}_{ho}^{\top}\mathbf{h}_{(t-1)} + \mathbf{b}_o\right) \tag{9}$$

$$\mathbf{g}_{(t)} \quad = \tanh\left(\mathbf{W}_{xg}^{\top}\mathbf{x}_{(t)} + \mathbf{W}_{hg}^{\top}\mathbf{h}_{(t-1)} + \mathbf{b}_g\right) \tag{10}$$

$$\mathbf{c}_{(t)} \quad = \mathbf{f}_{(t)} \otimes \mathbf{c}_{(t-1)} + \mathbf{i}(t) \otimes \mathbf{g}_{(t)} \tag{11}$$

$$\mathbf{y}_{(t)} \quad = \mathbf{h}_{(t)} = \mathbf{o}_{(t)} \otimes \tanh\left(\mathbf{c}_{(t)}\right) \tag{12}$$

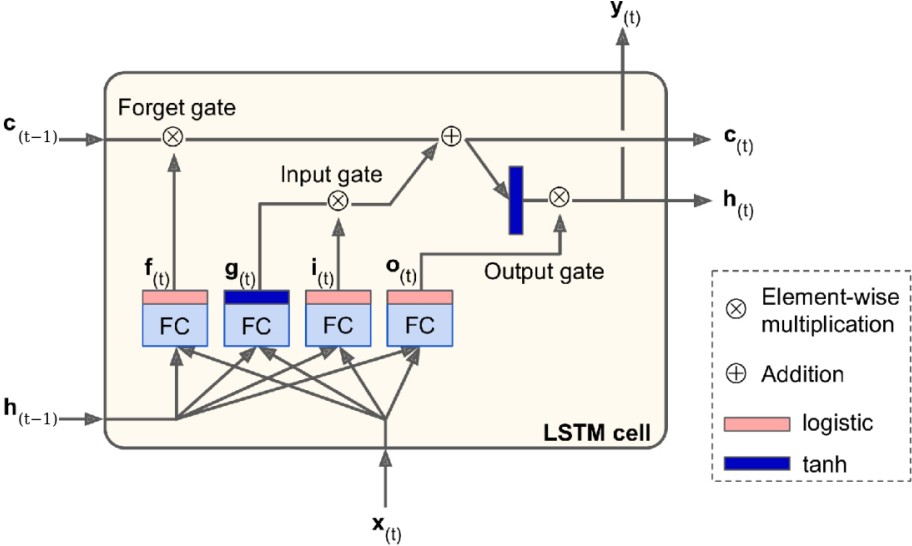

**Figure 4.** The LSTM cell improves upon the basic recurrent neural network with a series of gates to selectively remember and forget information passing through. FC stands for fully connected layer, which is internal to the cell, not the layers in Figure 3.

The four weight matrices are each $\mathbf{W}$, where the first subscript denotes the fully connected network input source, $\mathbf{h}_{(t-1)}$ or $\mathbf{x}_{(t)}$, and the second subscript denotes the control signal created: $\mathbf{f}_{(t)}$, $\mathbf{i}_{(t)}$, $\mathbf{g}_{(t)}$, or $\mathbf{o}_{(t)}$. Each of the four bias vectors is noted by $\mathbf{b}$, where the subscript denotes the control signal created.

An example of the full LSTM prediction model is shown with two features, two hidden layers, and an output ANN layer in Figure 5. The feature sequences have a length $S$, beginning at time $t - S$, and ending at time $t = 0$ which is the most recent load measurement. The features are processed chronologically by the LSTM layers, and then the single ANN neuron reduces the last hidden state to a single target, or output, at time $t + 24$ h. The hidden states have a length of $U$, called the number of "units". As the units increase so do the number of weights and biases, however for each hidden layer there is still only one set of $\mathbf{W}$-weights and (e.g., $\mathbf{W}_{xi}$, $\mathbf{W}_{hi}$) and one set of $\mathbf{b}$-biases (e.g., $\mathbf{b}_i$, $\mathbf{b}_f$). This is different from ANN networks where each neuron has its own $\mathbf{W}$ and $\mathbf{b}$ separate from other neurons. There is some ambiguity on this point in the literature.

Training is performed with the Adam variable learning rate optimizer for its efficiency and stability regarding non-stationary data [55], MSE for model loss, and a maximum batch size of 32. Training continues until the validation subset loss has not decreased in 100 epochs, a value which was experimentally found to be substantially high that the Adam optimizer finds a minimum without substantial overfitting.

**Feature 1**

**Figure 5.** In models with multiple hidden LSTM layers, the LSTM cell is connected to a second LSTM cell, and finally a ANN output layer resulting in a single predicted target at $t + 24$ h and features as shown. Multiple features are input to the same first hidden layer. The first LSTM layer has a set of weights $\mathbf{W_1}$ and $\mathbf{b_1}$, whereas the second LSTM layer has different weights, $\mathbf{W_2}$ and $\mathbf{b_2}$. The LSTM cell output, $\mathbf{y} = \mathbf{h}$ has the length $U$. The feature vectors and LSTM cells are tilted 90° clockwise for compactness.

*2.5. Hyperparameter Tuning*

The weights and biases of a machine learning model are set by the optimizer during training. Hyperparameters, instead, are the larger dimensions or characteristics of the model like the number of layers and size of the hidden state. The interactions between hyperparameters are poorly understood, so often a grid or randomized search is used to find the most optimal configuration. But as models become deeper and datasets larger the limitations of randomized search are quickly reached as it is computationally very expensive. The authors in [39] have developed Hyperband, an efficient hyperparameter tuning algorithm based on successive halving of the number of models in the search space, always preserving the best-performing half. Hyperband converges faster than Bayesian optimization on certain deep learning tasks, which in turn converges faster than randomized search [39].

The hyperparameter search space in the methodology includes the number of layers, equal to 3 or 4; dropout factors of 0, 10%, and 20%; one or two hidden layers; and LSTM units ranging from 24 to 1024. Dropout regularization helps prevent neural networks from overfitting, memorizing training data rather than learning the underlying patterns, by randomly setting to zero a chosen percentage of neurons during each training step to inject some noise and thereby mitigate redundant (co-adapted) learning among layers [45]. Units refer to the dimension of the LSTM cell state $\mathbf{c}$, as well as the output $\mathbf{y}$ and hidden state $\mathbf{h}$ which are the same signal.

*2.6. Computational Burden Estimation*

One feature of the methodology is to reduce the computational burden common to very large and deep models. Since the time and power required to train a model varies by hardware and software, the total number of model weights is a helpful proxy. Given that $\mathbf{x}_{(t)}$ must be a vector with a length equal to the number of features $f$, and $\mathbf{h}_{(t)}$ must be a vector with length equal to the number of units $u$. Then we can deduce that the all the $\mathbf{W}_x$ matrices ($\mathbf{W}_{x,i}$,.. etc) must have a shape $[u \times f]$, all the $\mathbf{W}_h$ matrices ($\mathbf{W}_{h,i}$,.. etc) must have a shape $[u \times u]$, and all the bias vectors $\mathbf{b}$ ($\mathbf{b}_i$,.. etc) must have a shape $[u \times 1]$. Therefore the simplified expression for the number of parameters (weights and biases) in the first hidden layer, $p_{H1}$, must be:

$$p_{H1} = 4u(u + f + 1) \tag{13}$$

However, the next hidden layer has a different input vector, because that layer takes the hidden state $\mathbf{h}$ as an input. Therefore $u$ replaces $f$ and the number of parameters in any second hidden LSTM layer, $p_{H2}$, must be:

$$p_{H2} = 4u(2u + 1) \qquad (14)$$

Finally the last hidden layer's hidden state is fully connected to a dense output layer to produce a single forecasted value. The number of output layer parameters, $p_O$, is equal to the length of the hidden state plus one bias value:

$$p_O = u + 1 \qquad (15)$$

### 3. Case Studies

The proposed methodology is applied to several different datasets, summarized in Table 2. All measurements are real power averaged over an interval of 10, 15, or 60 min. Most of the datasets have between 100,000 and 150,000 samples except for Hotel 2 which only has 70,000. The Residence, Transmission Network, and EV Charging Station datasets are available online at the links provided in the Data Availability Statement at the end of this work. Autocorrelation is calculated to guide the selection of the best naive persistence lag value, either 1 or 7 days.

An example of the cleaned load data is shown for Hotel 1 in Figure 6, downsampled to a daily average for viewing. A strong seasonality and slightly decreasing trend can be seen, suggesting non-stationarity. Most of the datasets show modest weekly and annual seasonality, whereas the Residence exhibits a strong trend over time.

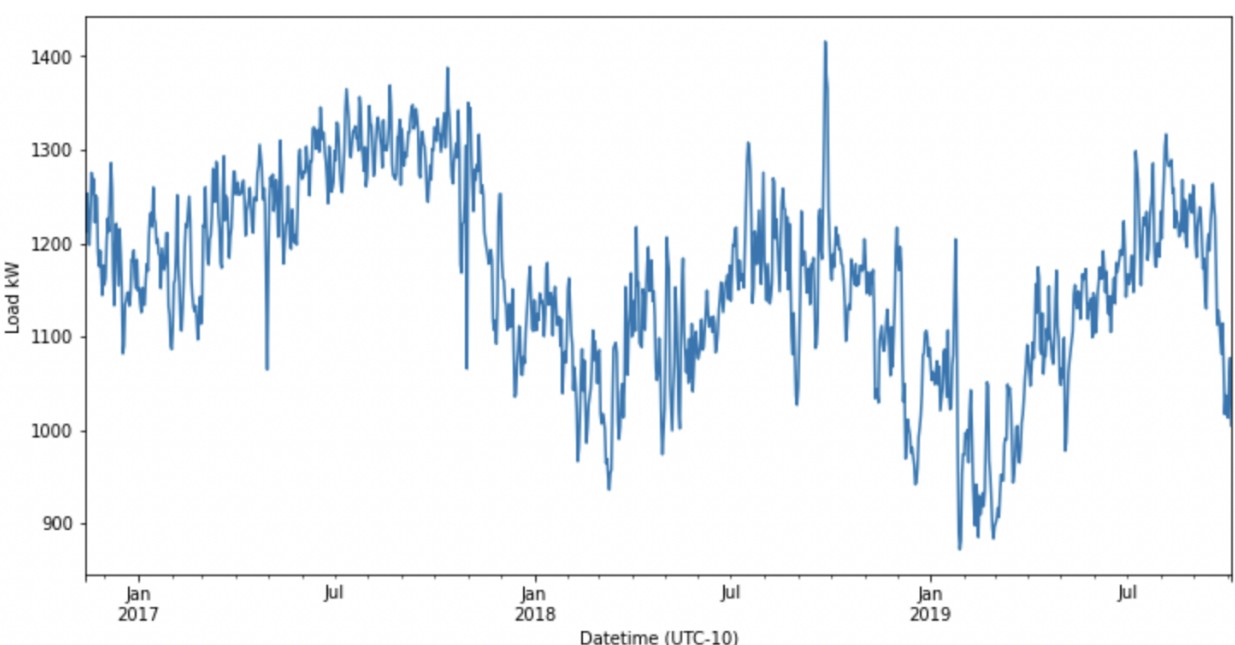

**Figure 6.** Hotel 1 load cleaned of outliers and resampled (for viewing) with a daily average, a strong seasonality and slight decreasing trend can be seen.

The methodology was developed on the Hotel 1 data, whereas Hotel 2 is intended as a test dataset from the same load sector. This informs how well the model generalizes within the same load sector. The Residence dataset is important because it is publicly available for other researchers to compare against. Single residences often have a very high degree of load disaggregation and the consumption patterns are driven by less predictable human behavior, indeed the seven-day autocorrelation is 0.63, the second lowest among those in Table 2. The Manufacturing Plant dataset is interesting because the weekend behavior is very different from weekdays, based on an autocorrelation that is much higher for a seven-day lag than one day. This concept drift will likely also occur on holidays and can affect the forecast performance greatly. The EV Charging Station represents the most

disaggregated load. While there are 54 charging points, typically less than ten EVs are charging at any given time. However is not necessarily the least predictable, considering a seven-day autocorrelation of 0.84 and good stationarity. The focus of this work is on building-scale load, but the Distribution Network and Transmission Network (distinct, unrelated datasets) are included for comparison of spatial aggregation and due to the well-studied nature of transmission network load.

The methodology is implemented with Keras and Tensorflow 2.6.0 on Google Colaboratory virtual machines with 2 GB of GPU RAM and 16 GB of CPU RAM. Keras and Tensorflow are the chosen libraries due to the high level of customization, good documentation, and good integration with Colaboratory [56].

## 4. Results

Since raw measurements were obtained for the first four datasets, pre-processing is an important part of the methodology that is obscured by the final error metric. Therefore these intermediate results are discussed first.

### 4.1. Data Pre-Processing

Even when limiting subset size to 5000, Shapiro-Wilk $p$-values on subsets of the different load datasets are rarely above an $\alpha$ of 0.05. The most Gaussian dataset is Hotel 1, seen in Figure 7 to still exhibit some leftward skew. However, z-score is still a reasonable method to select outliers because it is calculated based on the standard deviation of measurements at the same hour of each day. For these subsets, $p$-values range from 0 to 0.3, with most between 0.1 and 0.2. For all Hotel 1 values in the 09:00 to 10:00 period, the $p$-value is 0.118, and across the datasets, the probability distributions of values from the same hour of the day are much more Gaussian than the entire dataset.

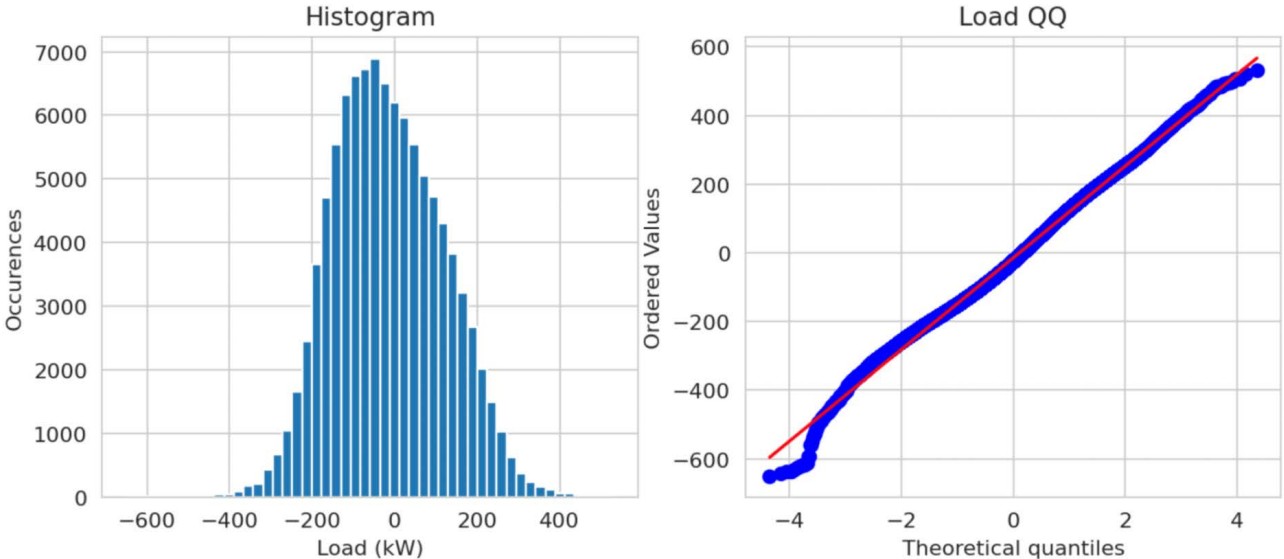

**Figure 7.** Hotel 1 data are almost normally distributed but with some leftward skew. The Quantile-Quantile or Q-Q plot (right) shows which measurement quantile matches or deviates from the theoretical quantile. In this case, the Q-Q plot shows deviation from normality in the left and right tails, and also in the second-from-left quartile.

Stationarity is mostly true for the datasets based on the ADF statistic being well below the 1% threshold statistic, and the $p$-value below an $\alpha$ of 0.05. The Hotel 2 and Distribution Network had a slight trend over time, whereas the Residence has a strong upward trend.

Outlier detection often found null-value measurements, statistical outliers, and segments of data that should be imputed or excluded from the forecast for strategic reasons. In Figure 8 two outlier measurements are labeled on 12 December of the Hotel 1 dataset.

The first is labeled by z-score because the measurement is outside of three standard deviations from the mean of the hour-weekday subset of Monday 1:00–2:00. The second is labeled by k-means clustering as being geometrically closer to the outlier than the centroid of the nearby points.

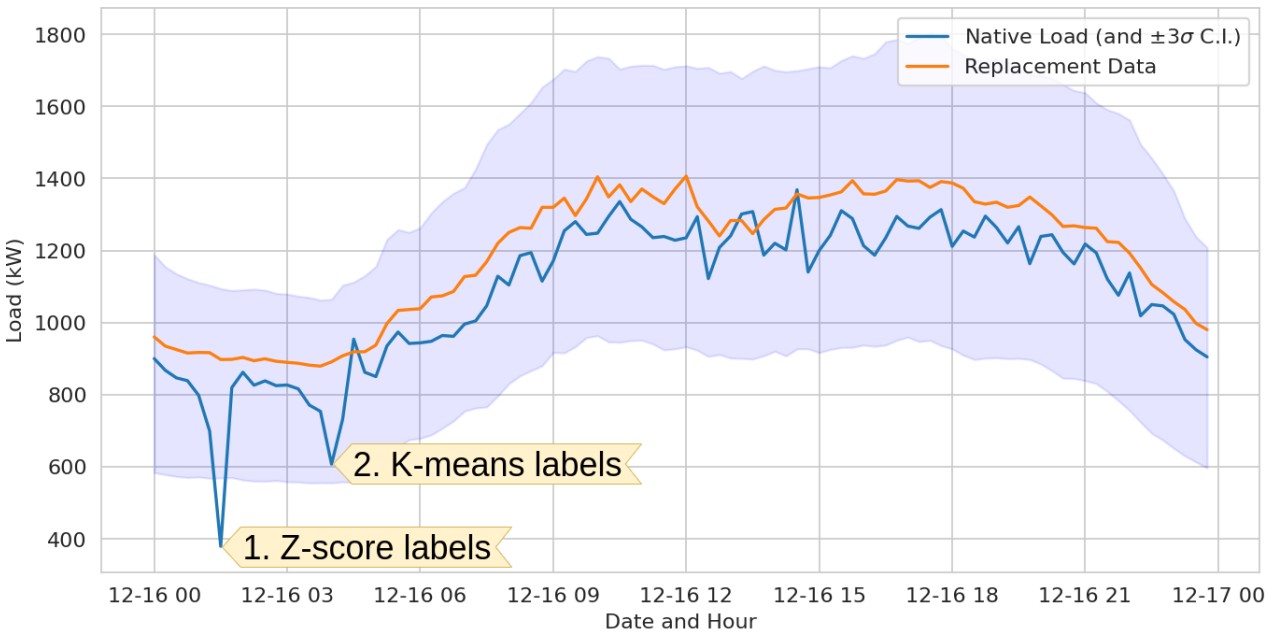

**Figure 8.** Z-score labels statistical outliers, for instance (1) is more than three standard deviations from the mean of all measurements from 1:00–1:45 on Sundays. Then k-means clustering labels geometrically close points, such as (2). The two outlier measurements are replaced with the corresponding values in Replacement Data, which come from 7-day naive persistence according to Table 2. Only the two data points are replaced (the Replacement Data curve is shown for reference).

For example, with the Hotel 1 dataset, the z-score and k-means combined method labeled 8 months of low load-magnitude which had very little daily seasonality, likely due to low occupancy and limited heating and cooling during COVID-19 health restrictions. A period of bad measurements were identified which seemed to occur every day at sunrise, possibly a metering error when a solar photovoltaic installation was completed. This resulted in decreasing the dataset by 26%. Additionally, 0.3% (492 measurements) of the data were identified as statistical outliers.

The EMD algorithm produced between 10 and 12 IMFs per dataset, which correlated with the endogenous load between 0.05 (often IMF9 and IMF10) and 0.7 (IMF2 or IMF4). Those with correlation coefficients above 0.35 were added as features, a threshold established after the Hyberband tuning algorithm was consistently selecting those IMFs.

A good example of the resultant IMFs is in Figure 9 from Hotel 1. The last two IMFs represent the overall trend and annual seasonality. The IMFs 3–5 represent daily and intra-daily periodicity. IMFs 3 and 4 have Pearson correlations of 0.50 and 0.73 concerning the original load data. The statistical properties of the IMFs differ: IMFs 1–5 are somewhat Gaussian and IMFs 10–11 are extremely non-Gaussian.

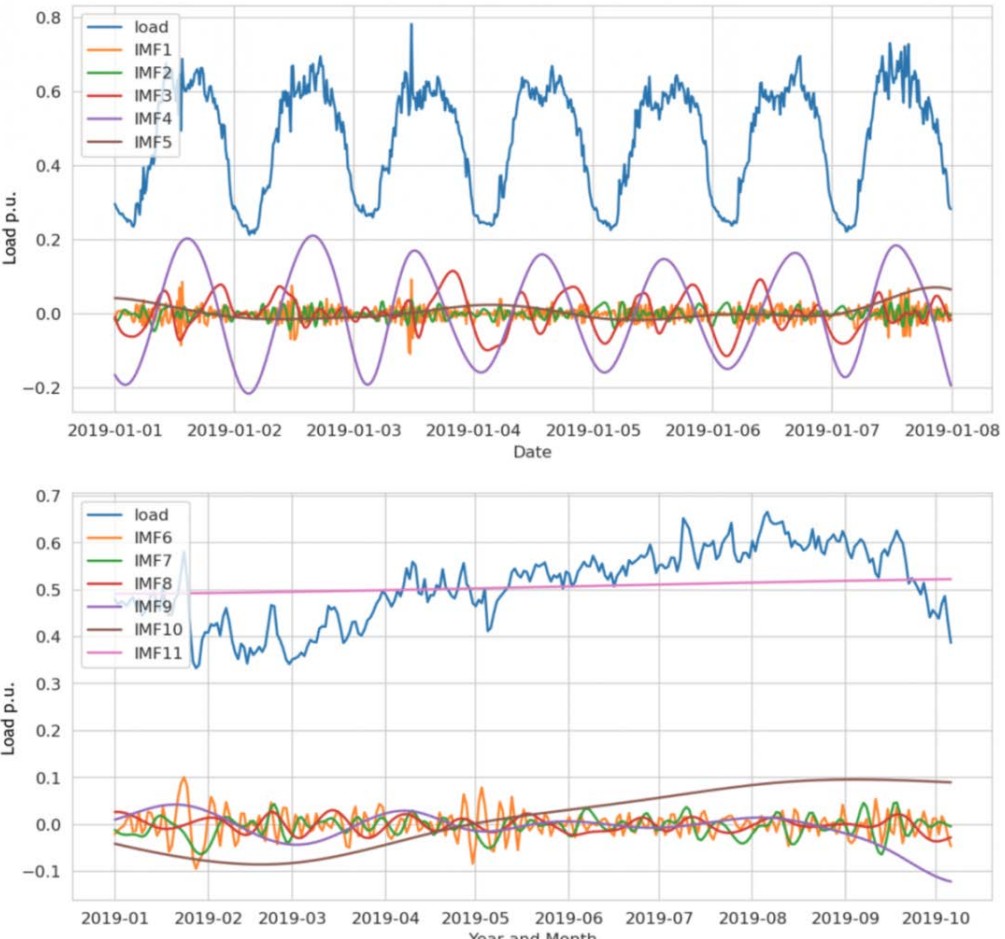

**Figure 9.** EMD performed on the Hotel 1 load which produced 11 IMFs. This image should be viewed in color for convenience.

*4.2. Forecast*

An example period of predicted and meausured data is shown in Figure 10 from Hotel 1. For each dataset, the best model is presented in Table 3 with its hyperparameters and test set error, which is the last 10% of the data. Cases ending in b (e.g., 1b) show the improvement over Cases ending in a when the EMD IMFs are included. The number of weights approximates the complexity and size of the model. The benchmark models seasonal NP and SARIMA are compared via skill score. SARIMA is trained using the Python pmdarima package which automatically performs a small grid search on the training data. Although cases 5 and 6 in Table 3 might appear to be somehow mixed, they are indeed separate case studies that have no impact on each other. Each is included in this paper because the datasets are very long compared to the other case studies except the Residence, and an expected result is that the LSTM model will better learn patterns if there is more data. This does not seem to be exclusively the case. Note that the Manufacturing Plant and Distribution Network have similarly high autocorrelation values (0.93 and 0.94 respectively, Table 2) but even with more than three times the data the LSTM-EMD skill (relative to SARIMA) is substantially lower for the Distribution Network than the Manufacturing Plant. Since the opposite is true of the LSTM-EMD skill relative to NP, a possible explanation is that the Distribution data is less stationary based on the ADF statistic in Table 2.

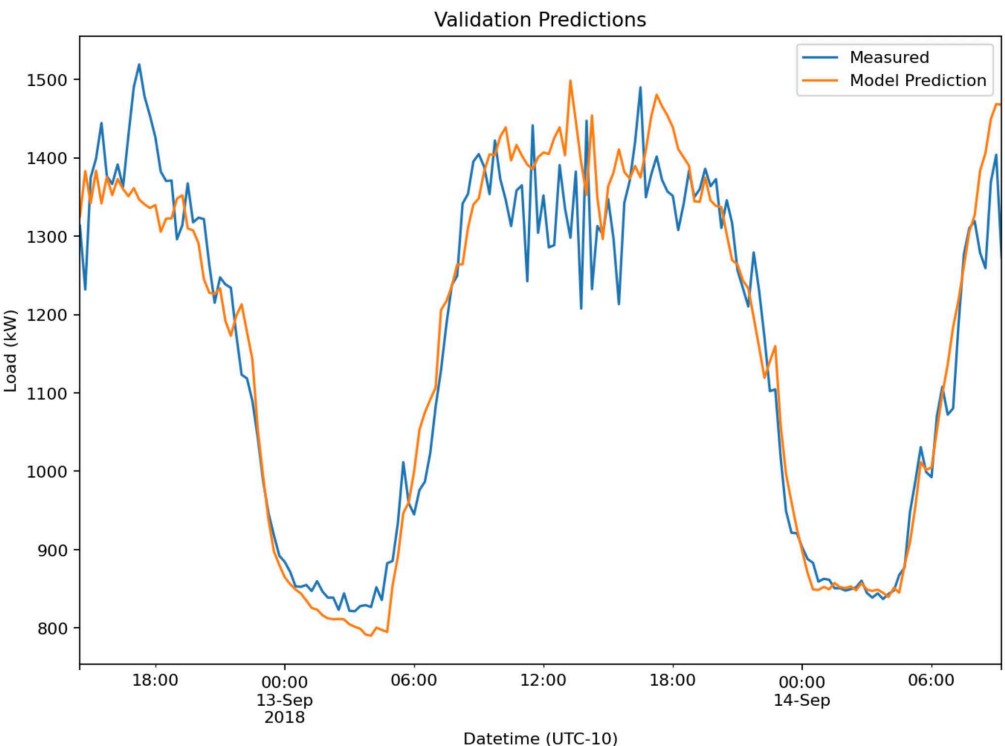

**Figure 10.** LSTM-EMD forecast model is qualitatively shown to capture some of the non-linearity and high variance, in this case during midday for the Hotel 1 data. The high midday variance in midday measurements and therefore forecast errors is likely due to the hotel operations which are disaggregated (large cooling compressors and fans) and time-ambivalent occupant behavior (compared to morning and evening which are likely more fixed by staff schedules).

**Table 3.** Results Summary.

| Case | Site | LSTM Hyperparameters IMFs | Units | Dropout | Layers | Weights [×10⁶] | Test Error [RMSE] | NP Test Error [RMSE] | [SS] | SARIMA Test Error [RMSE] | [SS] |
|------|------|------|-------|---------|--------|----------------|-------------------|------|------|------|------|
| 1a | Residence | – | 1024 | 0.1 | 3 | 4.215 | 0.71 kW | 0.73 kW | 2.7% | 0.70 kW | −1.4% |
| 1b | | 1,2,3 | 256 | 0 | 3 | 0.280 | 0.63 kW | | 14% | | 10% |
| 2a | Hotel 1 | – | 256 | 0 | 4 | 0.793 | 62.41 kW | 73.05 kW | 15% | 56.5 kW | −11% |
| 2b | | 3,4,5 | 256 | 0 | 3 | 0.271 | 60.05 kW | | 18% | | −6.3% |
| 3a | Hotel 2 | – | 128 | 0 | 3 | 0.068 | 25.27 kW | 33.90 kW | 25% | 24.4 kW | −3.6% |
| 3b | | 3,4,5 | 512 | 0 | 4 | 3.165 | 24.34 kW | | 28% | | 0.2% |
| 4a | Manufacturing Plant | – | 96 | 0 | 4 | 0.113 | 598.7 kW | 595.1 kW | −0.6% | 1697 kW | 65% |
| 4b | | 3,4,5 | 96 | 0 | 4 | 0.114 | 451.1 kW | | 24% | | 73% |
| 5a | Distribution Network | – | 128 | 0.1 | 4 | 0.200 | 51.01 MW | 58.18 MW | 12% | 58.1 MW | 12% |
| 5b | | 3,4,5 | 96 | 0.1 | 3 | 0.040 | 39.40 MW | | 32% | | 32% |
| 6a | Transmission Network | – | 512 | 0.1 | 4 | 3.159 | 1952 MW | 3594 MW | 46% | 4236 MW | 54% |
| 6b | | 3,4,5 | 512 | 0.1 | 4 | 3.165 | 1580 MW | | 56% | | 63% |
| 7a | EV Charging Station | – | 96 | 0 | 4 | 0.0039 | 11.23 kW | 24.27 kW | 54% | 19.6 kW | 43% |
| 7b | | 3,4,5,6 | 512 | 0.1 | 3 | 1.067 | 8.93 kW | | 63% | | 54% |

LSTM training was typically stable, with model loss (MSE) decreasing monotonically except for occasional explosions in the gradient followed by a return to a similar loss magnitude. Validation subset loss was similar to test subset loss for the transmission systems, but for the single buildings often would drop quickly to a plateau, occasionally experiencing an exploding gradient and then returning to the plateau.

Of the four models with over a million weights, three benefitted from dropout presumably to avoid some overfitting. An advantage of Hyperband is that it is more resource efficient so more values of hyperparameters can be tested. One best model utilized the maximum number of 1024 units but none of the solutions had the lower values of 24 or 48 units.

Based on these results, a minimum threshold of 0.9 is suggested for data with good autocorrelation. For those datasets, the NP RMSE normalized by peak load is 4.1%, 4.6%, and 3.1% for the Hotel 1, Manufacturing Plant, and Distribution Networks, respectively. An expected and positive result is that the respective skill score for each of those three datasets was 18%, 24%, and 32%, so the highest skill score is associated with the lowest NP RMSE. This is to say that the most predictable dataset according to NP was also the dataset where the LSTM-EMD model was able to make the most improvement over NP. A possible explanation is that spatial load aggregation benefits the Distribution network the most, in terms of prediction capability. The Manufacturing Plant is physically large enough to have good load aggregation, but as the building is a manufacturing production line many of the individual machines will be temporally aligned.

Skill score in Table 3 is generally favorable for the models with IMF features, ranging from 18% (Hotel 1) to 63% (Transmission Network) for NP and −6.3% (Hotel 1) to 73% (Manufacturing Plant) for SARIMA. SARIMA outperforms or reaches parity with LSTM-EMD on both hotel datasets, which is an important finding considering that the hotels are of roughly similar size and operation. A hypothesis is that the hotels are the two most consistent datasets causing SARIMA to perform better with respect to LSTM-EMD. Each hotel has a fairly high autocorrelation value of 0.95 and 0.87 for one-day seasonality. The two hotels are also nearly Gaussian distributions. Although both also have the two highest ADF statistics of −5.5 and −4.9, all the datasets are below the 1% critical value of −3.4. This is a very positive finding but is difficult to compare against other works in the literature where the error metrics, forecast period, models, data, and benchmarks are all somewhat different. Also notable is that skill score always increases with the addition of IMFs as features, from +3% to +25%. However model complexity (estimated as the number of weights) remained constant or increased in five of seven datasets, so the feature extraction may not greatly reduce the computational burden. This isn't necessarily surprising since test set RMSE wasn't held constant. In only one case was there a negative skill score, the Manufacturing Plant without EMD only did as well as −0.6%. Since EMD is formally part of the methodology this result merely indicates that even modern machine learning techniques don't always beat a single parameter NP model when the lag $L$ is well-chosen.

## 5. Conclusions

In this work an LSTM EMD day ahead load forecast methodology is tested on seven diverse datasets from six different load sectors. The forecasted test set RMSE is lower than the benchmark models in six out of seven case studies, ranging from −6.3% to 73% improvement over the benchmarks. The two hotel data sets have the worst LSTM-EMD performance relative to the SARIMA benchmark, which could be due to each time series having higher autocorrelation values at their first seasonality lag (one day) and being close to normally distributed, relatively favoring SARIMA given the quantity of data. None of the datasets meet the $p = 0.05$ threshold for normality using the Shapiro-Wilk test, though all have an ADF test statistic below the 1% critical value of −3.4. Autocorrelation values range from 0.45 to 0.94, for lag values of one and seven days. Confidence in the forecast results is high given that separate validation and test sets were used, dropout is included to help prevent overfitting, and model loss during training generally decreased monotonically. In some exceptions, the error gradient temporarily exploded, especially on datasets with higher time resolution. The LSTM cell is intended to mitigate this problem, but not eliminate it. Adding IMFs as features convincingly increases skill score, which is a very positive finding. Future work should include many building samples for each load

sector, consider deeper and more complex model structures, and include online learning to adapt to concept drift.

**Author Contributions:** Conceptualization, M.W., E.O., A.N., S.L.; data curating, M.W., A.N.; formal analysis, M.W.; funding acquisition, T.S.; investigation, M.W., A.N., E.O.; methodology, M.W., A.N., E.O., S.L.; project administration, E.O., S.L.; resources, M.W., E.O.; software, M.W.; supervision, E.O., T.S., S.L.; validation, M.W., E.O., A.N., S.L.; visualization, M.W.; writing—original draft, M.W.; writing—reviewing and editing, M.W., E.O., A.N., T.S., S.L. All authors have read and agreed to the published version of the manuscript.

**Funding:** This research received no external funding.

**Data Availability Statement:** The raw data required to reproduce the above findings in the Residential Cases 1a and 1b (Table 3) are available to download from https://dx.doi.org/10.17041/1880667, accessed on 23 August 2022. The raw data for the Transmission Network Cases 6a and 6b are available to download from https://www.terna.it/en/electric-system/transparency-report, accessed on 2 July 2022, but without a permanent link. The raw data for the EV Charging Station [57] Cases 7a and 7b are available to download from https://doi.org/10.1145/3307772.3328313 accessed on 5 September 2022. The raw data for the remaining cases are the ownership of private entities and are used with their permission but cannot be distributed.

**Acknowledgments:** The authors warmly thank muGrid Analytics for a few of the datasets and information about the business context of the distributed energy optimization problem.

**Conflicts of Interest:** The authors have no conflicts of interest to report.

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
