# Peer review of "Day Ahead Electric Load Forecast: A Comprehensive LSTM-EMD Methodology and Several Diverse Case Studies"

_forecasting, doi:10.3390/forecast5010016_

Round 1

Reviewer 1 Report

This paper presents a proposal with EMD and LSTM for forecasting data from different load sectors.

For me, many basic concepts are presented and do not add any value to the paper (e.g. section 2.2).

Although several datasets were used, and the results are promising, the innovative contribution is unclear to me.

What do the authors add to the literature with a pre-processing, filtering (EMD) and forecasting (LSTM) sequence of steps?

Where is the turning point that characterises innovation? Would it be the Hyperparameter Tuning and the Computational Burden Estimation steps? Please, clarify.

Regarding reproducibility, are the codes available?

It would be helpful to show the graphs of all datasets. Further discussions about Covid-19 impacts are recommended. 

Reviewer 2 Report

The manuscript presents an interesting methodology embracing Long Short-Term Memory and Empirical Mode Decomposition to predict day-ahead electric load. The methodology is applied to different case studies, involving considerably distinct spatial resolution and diverse activity sectors (diverse consumption aggregation levels).

The manuscript is very well organized, covering in a fair and in a suitable way the main topics related with the proposed approach. The presented methodology and the analysis are carefully described by the authors with critical judgments.

Nevertheless, some suggestions are proposed to the authors to improve the quality of paper:

-          Fig. 1 with the process diagram: despite some of them have already presented before, used abbreviations should be declared in the figure (like NP, ACF and IMF). For instance, IMF abbreviation is only described in Page 6.

-          More space should be included in-between Table 1 and Figure 1.

-          In the methodology section, it should be clearly described which resolution is being explored along the article: to predict day-ahead hourly load values or other resolutions different from 1 hour?

-          In Section 2.2.3, when Equation 5 related with Augmented-Dicky-Fuller test is described, it should be better described the meaning of the terms involving deltas.

-          In Section 2.3, it should be better justified why a triangle shaped feature is being used to characterize the day-of-year, day-of-week and hour-of-day. Being a cyclic variable, couldn’t it be converted by using a sinusoidal function?

-          In Section 2.4 several details related with hyperparameter tuning are described. In this section, some LSTM specificities (such as dropout factors or number of LSTM units) are pointed out.  However, in my opinion, this should be described only after a better presentation of LSTM model (an effort to change the order between Sections 2.4 and 2.5 without losing the context should be considered).

-          It should be reconsidered the order of Figures 3 and 4. As the main text describes firstly Figure 4, (by using a cross-reference), perhaps it should appear before.

-          Figure 3 caption describes some lack of transparency in the literature about the concatenation procedure inside the LSTM cell. A better description and some reflection on this scientific void should be inserted in the main text.

-          It seems that authors have used Tensorflow to all methodologies tested. Is it possible to give some advantages of this library when compared with other (like Scikitlearn,…). Is the main advantage of Tensorflow its suitability for LSTM approaches?

-          In Table 2 autocorrelations associated with 1 and 7 days before are presented. As it is expected that non-linear dependences can be found between previous and predicted hourly values, when simply using an autocorrelation is there a risk of being simply looking for linear dependences and discarding eventual nonlinearities?

- in section 4.1, too much analysis are being presented for different case studies and, in my opinion, only based on text. To give a better perspective, I recommend an effort to synthesize information, by using Tables or Figures to support.

-          Fig. 8 shows the EMD performed on the Hotel 1. It is not well understood the vertical axis unit, because it is referred to “Load (kW/kW)”? Does this measuring unit have a practical meaning? Shouldn’t it be more a normalized power?

Reviewer 3 Report

In the manuscript, the authors use the proposed LSTM-EMD method to perform predictions and comparisons on seven different power system datasets. There are several suggestions for consideration.

1.The main contributions of the paper are clearly described. Nonetheless, the novelty of this work cannot be appreciated from the current manuscript. The authors should do a better job of highlighting the innovative aspects of their work in the manuscript.

2.To substantiate the effectiveness of the proposed method, the authors should emphasize the advantages over other state-of-the-art alternatives.

3.The LSTM method is currently widely used in forecasting in many fields, including Short-term load forecasting. The author can add literature descriptions and emphasize the significant differences with other load forecasting studies.

4.Could you explain why the Adam optimizer was chosen (line 294)? Have you tried other than 100 epochs?

Reviewer 4 Report

The results shows that the prediction accuracy for high loads is decreasing. Author to justify this observation. Also to explain more about the prediction model accuracy in case the peak load is increased to 200% percent of higher.

2. The author needs to compare the suggested model with other applicable models listed in his literature review.  

3. It is not clear why the author mix the data of the distribution load and transmission load in the moddeling?

Round 2

Reviewer 1 Report

The current version can be published.

Author Response

Thank you for the feedback, no additional changes have been made.

Reviewer 3 Report

the authors have completed improvements to these comments.

Author Response

(The authors gave the same response as above.)

Reviewer 4 Report

It is still not clear why the author mix the data of the distribution load and transmission load in the moddeling? Note in Table 3 and or the text related to Table 3 may be added to elaborate more to avoide confusion. 
